# MOSS: Efficient and Accurate FP8 LLM Training with Microscaling and Automatic Scaling

Yu Zhang[1],    Hui-Ling Zhen[2],    Mingxuan Yuan[2],    Bei Yu[1]
[1]Chinese University of Hong Kong    [2] Huawei Noah's Ark Lab

## Abstract

Training large language models with FP8 formats offers significant efficiency gains. However, the reduced numerical precision of FP8 poses challenges for stable and accurate training. Current frameworks preserve training performance using mixed-granularity quantization, i.e., applying per-group quantization for activations and per-tensor/block quantization for weights. While effective, per-group quantization requires scaling along the inner dimension of matrix multiplication, introducing additional dequantization overhead. Moreover, these frameworks often rely on just-in-time scaling to dynamically adjust scaling factors based on the current data distribution. However, this online quantization is inefficient for FP8 training, as it involves multiple memory reads and writes that negate the performance benefits of FP8. To overcome these limitations, we propose MOSS, a novel FP8 training framework that ensures both efficiency and numerical stability. MOSS introduces two key innovations: (1) a two-level microscaling strategy for quantizing sensitive activations, which balances precision and dequantization cost by combining a high-precision global scale with compact, power-of-two local scales; and (2) automatic scaling for weights in linear layers, which eliminates the need for costly max-reduction operations by predicting and adjusting scaling factors during training. Leveraging these techniques, MOSS enables efficient FP8 training of a 7B parameter model, achieving performance comparable to the BF16 baseline while achieving up to 34% higher training throughput.

## 1    Introduction

Large language models (LLMs) have demonstrated remarkable capabilities across diverse tasks, including reasoning, language understanding, and generation (Achiam et al., 2023; Grattafiori et al., 2024; Liu et al., 2024; Adler et al., 2024). However, training these models, which often comprise billions of parameters, incurs substantial computational costs. By leveraging tensor quantization in deep neural networks, low-precision training has emerged as a promising solution to accelerate computation and reduce memory requirements. Among existing methods, BF16 training is currently the most widely adopted low-precision approach, supported by major large-scale training frameworks such as DeepSpeed (Rasley et al., 2020) and Megatron-LM (Shoeybi et al., 2019).

With recent advancements in hardware support for FP8 data formats, FP8-based low-precision training (Micikevicius et al., 2022) has emerged as the next-generation training technique. FP8 computation offers significant advantages over BF16 training, including up to $2\times$ speed-up, 50% reduced memory footprint, and 50% lower communication overhead in theory. However, the limited range and resolution of FP8 introduce substantial challenges for LLM training stability and convergence. To achieve practical benefits, Transformer Engine (Nvidia, 2025) first applies FP8 for general matrix multiplication (GEMM) while maintaining master weights and gradients in higher precision (FP16/FP32) to preserve training stability. FP8-LM (Peng et al., 2023) extends this by further quantizing gradients and first-order momentum to FP8, achieving additional memory savings. Due to FP8's sensitivity to activation outliers, recent studies have proposed fine-grained quantization strategies to enhance training stability. Notably, COAT (Xi et al., 2024) and DeepSeek-V3 (Liu et al., 2024) adopt per-group activation quantization to preserve numerical stability, while employing per-tensor/block weight quantization to minimize memory overhead.

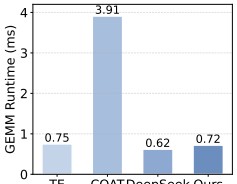

Figure 1: Quantized GEMM Runtime Comparison on H800.

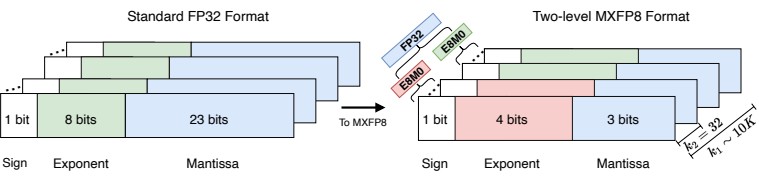

Figure 2: Depiction of MXFP8 data format with level-1 FP32 scale factor and level-2 E8M0 scale factor, with group size $k_1 \sim 10K$ and $k_2 = 32$ over which these two scaling factors are shared.

However, despite the high fidelity, existing per-group FP8 quantization schemes apply scaling factors along the inner dimension (K) of GEMM executed on Tensor Cores, which incurs additional dequantization overhead on CUDA Cores. As illustrated in Figure 1, the per-group quantized COAT (Xi et al., 2024) significantly underperforms compared to the per-tensor approach used in Transformer Engine (Nvidia, 2025), due to significant dequantization and partial sum overhead within the main computation loop. Additionally, dynamically computing per-tensor scaling factors for weights introduces memory access and max-reduction overhead during training, diminishing the benefits of FP8 training.

In this work, we present MOSS, a framework for efficient and accurate FP8 training that addresses the aforementioned limitations. For sensitive activations, MOSS introduces a progressive two-level quantization strategy: a global per-tensor scale in FP32 precision, combined with local microscaling using power-of-two factors. This design enhances numerical fidelity while reducing dequantization overhead in the GEMM kernel. For weights well-suited to per-tensor quantization, MOSS exploits the bounded update property of Adam-like optimizers (Kingma & Ba, 2014), where weight updates are inherently limited by the learning rate. Building on this, MOSS proposes an automatic scaling mechanism that predicts scaling factor evolution based on optimizer hyperparameters, thereby eliminating runtime max-reduction and its associated cost. Together, these innovations make MOSS a more efficient quantization framework for FP8 training, better suited to Tensor Cores while preserving training stability and accuracy.

We evaluate MOSS by training large language models, including OLMo-7B (Groeneveld et al., 2024) and LLaMA-2-7B (Grattafiori et al., 2024), under GPU resource-constrained settings, specifically, using only 8 Hopper GPUs for full-scale pretraining. Notably, although Hopper GPUs do not natively support microscaled data formats, we enable this functionality by implementing custom GEMM kernels using Triton OpenAI (2025). Experimental results demonstrate that MOSS achieves lossless accuracy on both pretraining and fine-tuning tasks, matching the performance of BF16 baselines. In terms of efficiency, MOSS delivers a 1.34× end-to-end training speedup on OLMo-7B compared to BF16 and outperforms the state-of-the-art FP8 training framework, COAT, by 12.3%, significantly reducing the training cost of large-scale models. To further validate the efficiency of MOSS's FP8 GEMM operations, we conduct an ablation study comparing its runtime against other FP8 GEMM kernels, including DeepGEMM from DeepSeek-V3 (Liu et al., 2024), which confirms the superior efficiency of MOSS's GEMM kernel design.

## 2 Preliminaries

### 2.1 FP8 and Microscaling Data Format

In FP8, each tensor $\boldsymbol{X}$ is mapped to its lower-precision counterpart $Q_{\boldsymbol{X}}$ via $Q_{\boldsymbol{X}} = \lceil \boldsymbol{X}/s \rfloor$, where $s$ is the FP32 scaling factor, enforcing all values to fall within the dynamic range of the FP8 format. This often necessitates the use of the less precise E5M2 format for certain tensors, such as gradients and activations, to accommodate wider ranges Nvidia (2025). The microscaling (MX) data format, initially proposed in (Darvish Rouhani et al., 2023), has been standardized by the Open Compute Project (Bita et al., 2023) with support from Microsoft, Intel, NVIDIA, and others. The primary difference between standard FP8 and MXFP8 quantization lies in the granularity of scaling. In particular, MXFP8 assigns a distinct scaling factor to each block of 32 consecutive values, enabling all values to be quantized using the higher-precision E4M3 format. A second key difference is in the representation of scaling factors: FP8 stores them in full-precision FP32, whereas MXFP8 uses an 8-bit power-of-two format (E8M0), significantly reducing storage and computational overhead.

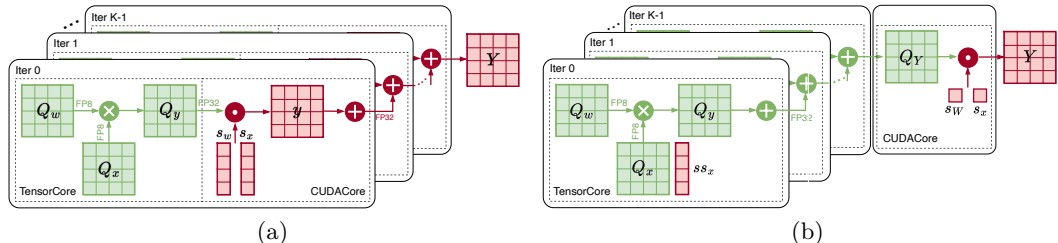

(a)            (b)

Figure 3: FP8 Quantized GEMM on GPUs. (a) Per-group FP8 GEMM in COAT suffers from significant dequantization overhead in the main loop. (b) In contrast, MOSS achieves faster matrix multiplication by confining the main loop to Tensor Core operations, with all dequantization deferred to the epilogue. Leveraging the fast MXFP8 GEMM module and a two-level microscaling quantization strategy, MOSS significantly reduces dequantization overhead and improves kernel efficiency.

## 2.2 Weight Update Rule

Optimizers play a crucial role in deep learning by updating model weights during training. The most common gradient-based optimizer is Adam/AdamW ( (Kingma & Ba, 2014; Loshchilov & Hutter, 2017)), which uses first-order $m$ and second-order momentum $v$ to achieve better convergence. Define $\nabla_{\mathbf{W}}\mathcal{L}(\mathbf{W}_t)$ as $\mathbf{g}_t$, the update rule for $l$-th layer at time step t using AdamW can be formulated as:

$$\mathbf{m}_t = \beta_1 \mathbf{m}_{t-1} + (1 - \beta_1)\mathbf{g}_t, \quad \hat{\mathbf{m}}_t = \frac{\mathbf{m}_t}{1 - \beta_1^t},$$

$$\mathbf{v}_t = \beta_2 \mathbf{v}_{t-1} + (1 - \beta_2)(\mathbf{g}_t)^2, \quad \hat{\mathbf{v}}_t = \frac{\mathbf{v}_t}{1 - \beta_2^t}, \tag{1}$$

$$\mathbf{W}_{t+1} = \mathbf{W}_t - \eta\left(\frac{\hat{\mathbf{m}}_t}{\sqrt{\hat{\mathbf{v}}_t} + \epsilon} + \lambda\mathbf{W}_t\right),$$

where $\mathbf{g}_t$ is the gradient, $\mathbf{m}_t$ is the first-order momentum, $\mathbf{v}_t$ is the second-order momentum, $\beta_1$ and $\beta_2$ are hyperparameters of AdamW, $\eta$ is the learning rate, $\lambda$ is weight decay, and $\epsilon$ (e.g., $1e-8$) is used to prevent NaN or Inf. A key advantage of Adam-like optimizers is that the magnitude of weight updates, $\Delta_t = \eta \cdot \hat{\mathbf{m}}_t/\sqrt{\hat{\mathbf{v}}_t}$, is invariant to diagonal rescaling of the gradients. Specifically, scaling the gradients $\mathbf{g}_t$ by a factor $s$ results in $\mathbf{m}_t$ scaling by $s$ and $\mathbf{v}_t$ by $s^2$, which cancels out in the update rule: $(s \cdot \hat{\mathbf{m}}_t)/\sqrt{s^2 \cdot \hat{\mathbf{v}}_t} = \hat{\mathbf{m}}_t/\sqrt{\hat{\mathbf{v}}_t}$. This scale-invariance contributes to the robustness of Adam-like optimizers in FP8 training, especially when handling FP8 gradients with high dynamic range.

## 3 Methods

### 3.1 Two-Level MicroScaling

Overview. Per-group activation quantization in current FP8 training frameworks (Xi et al., 2024) introduces a critical drawback: the insertion of per-group scaling factors along the inner dimension K of GEMM operations incurs significant dequantization overhead. This overhead is executed on slower CUDA Cores rather than high-throughput Tensor Cores (see GEMM kernel illustration in Figure 3a). As a result, the dequantization steps within the main loop of FP8 GEMM become a major efficiency bottleneck. Similar dequantization-induced CUDA Core overhead has also been observed in inference systems such as QServe (Lin et al., 2024). Specifically, on modern data center GPUs such as the NVIDIA H100, the peak throughput of FP32 CUDA Cores is only 1.6% of that of FP8 Tensor Cores. As a result, dequantizing even a single partial sum in a per-group quantized FP8 GEMM can cost the equivalent of nearly 60 Tensor Core multiply-accumulate (MAC) operations. This imbalance causes the GEMM main loop to be dominated by slow dequantization on CUDA Cores, effectively nullifying the performance advantages of FP8 training.

To strike a balance between quantization accuracy and GEMM efficiency, we propose a two-level microscaling approach that effectively approximates full-precision values with minimal overhead. As illustrated in Figure 2, MOSS hierarchically partitions a global block of size

$k_1$ into multiple equal-sized subpartitions of size $k_2$. A global scaling factor $s$ is applied uniformly across all partitions, while local sub-scale factors $ss_i$ are quantized for efficient storage and computation and applied to each corresponding $i$-th subpartition. In the first stage, we compute floating-point scale factors at a fine granularity. For the $i$-th partition with size $k_2$, $s_i$ is computed as

$$s_i = \max(|\boldsymbol{X_i}|)/\Delta_{\max}^{\mathrm{FP8}}, \text{where } i \in [0, k_1/k_2 - 1] \qquad (2)$$

where $s_i$ is the scaling factor of subpartition $i$, $\boldsymbol{X}_i$ is the $i$-th group vector and $\Delta_{\max}^{\mathrm{FP8}}$ is the maximum representable value in the chosen FP8 format. For common FP8 encodings (Micikevicius et al., 2023), $\Delta_{\max}^{\mathrm{E4M3}} = 448$ and $\Delta_{\max}^{\mathrm{E5M2}} = 57344$. In the second stage, we further quantize the per-group scale factors $s_i$ by separating them into E8M0 microscaling components $ss_i$ and a floating-point global component $s$,

$$s = \max(|\boldsymbol{s_i}|), \quad ss_i = \lceil s_i/s \rceil_{E8M0} = 2^{\lceil \log_2(s_i/s) \rceil}, \qquad (3)$$

where $\lceil \cdot \rceil_{E8M0}$ denotes rounding to the target data format E8M0, and $2^{\lceil \log_2(s_i/s) \rceil}$ denotes the smallest power of two that is greater than or equal to $s_i/s$. Dividing by $s$ helps shift the values toward the fractional range, making them more favorable for E8M0 quantization rounding and thereby improving quantization accuracy. In this way, the level-1 global scaling factor $s$ is encoded in full-precision FP32 for high accuracy, while the level-2 microscaling factors $ss_i$ are stored using an efficient 8-bit exponent-only format (E8M0), making them inexpensive to store and compute with. The dequantization operation that maps a quantized tensor back to FP32 can be expressed as: $DQ_{\boldsymbol{X_i}} = Q_{\boldsymbol{X_i}} \cdot s \cdot ss_i$. This two-level microscaling approach shifts the costlier FP32 scaling to a coarser granularity while introducing lightweight E8M0 scalers at a finer granularity, effectively balancing numerical accuracy with hardware efficiency.

Following the MX format specification (Bita et al., 2023), we set the level-2 group size to $k_2 = 32$. To support low-latency access within the inner loop of Tensor Core MMAs, the level-2 scaling factors $ss_i$ are stored in a contiguous memory layout aligned with the expected access patterns. This organization ensures efficient memory utilization and prevents stalls during the GEMM main loop. For model weights, we apply per-tensor quantization with FP32 scaling factors to fully exploit Tensor Core performance. The quantized matrix multiplication in Figure 3b is performed as: $Q_y = Q_w \times (Q_x * ss_x)$, where $Q_x$ is the FP8-quantized activation, scaled by E8M0-encoded level-2 factors $ss_x$, and $Q_w$ is the FP8-quantized weight tensor.

We illustrate our quantized GEMM kernel design in Figure 3b. To enable MXFP8 GEMM on Tensor Cores, we assign an artificial level-2 scaling factor with value 1 (encoded in E8M0) for $Q_w$, ensuring that GEMM operations between quantized activations and weights can be executed efficiently at fine granularity, balancing throughput and numerical precision. The resulting FP32 partial sum $Q_y$ is then dequantized in CUDA Cores using the FP32 per-tensor scale factor $s_W$ associated with the weight tensor and the level-1 FP32 global scale factor $s_x$ of activations (see CUDA Core in Figure 3b). Compared to traditional per-group quantized GEMMs, this two-level quantization scheme maintains high fidelity while significantly reducing dequantization overhead in the GEMM main loop.

Quantization Precision Analysis. In the numerical precision analysis for quantization methods, we use Signal to Noise Ratio (SNR), which measures the ratio of the original signal's power to the noise power introduced by quantization. The quantization SNR is calculated as,

$$\mathrm{SNR} := \frac{P_{\mathrm{signal}}}{P_{\mathrm{noise}}} = 10 \log\left(\frac{\mathbf{E}[\|\boldsymbol{X}\|^2]}{\mathbf{E}[\|\boldsymbol{DQ_X} - \boldsymbol{X}\|^2]}\right), \qquad (4)$$

where $\| \cdot \|$ denotes the root mean square (RMS) norm, defined as $\|\boldsymbol{X}\| = \sqrt{\sum_{i=1}^{k} x_i^2}$. A higher SNR indicates better quantization fidelity, as it reflects lower noise relative to the original signal. In the context of language model quantization, various schemes are used to balance precision and efficiency. Beyond our proposed two-level microscaling in MOSS, common approaches include per-tensor quantization and per-group quantization. In per-tensor quantization, a single scaling factor is shared across the entire tensor, resulting in minimal overhead but limited adaptability to local value variations. In contrast, per-group quantization partitions the tensor into sub-groups along channels, each with its own scaling factor, improving accuracy at the cost of increased metadata and computational overhead.

We then present a theoretical analysis of the quantization precision for per-tensor and per-group approaches, as well as our proposed MOSS framework.

Theorem 1. The quantization SNR of per-tensor, per-group quantization and two-level microscaling in MOSS follows: $\text{SNR}_{\text{per-tensor}} < \text{SNR}_{\text{per-group}} < \text{SNR}_{\text{MOSS}}$.

Proof 1. In the statistical analysis, we assume without loss of generality that the elements of $\boldsymbol{X}$ are zero-mean. Under this assumption, the signal power is given by $P_{\text{signal}} = \sigma_X^2$, where $\sigma_X$ denotes the standard deviation of $\boldsymbol{X}$. First, given that floating-point quantization belongs to uniform quantization, the per-tensor quantization error $e$ is uniformly distributed in the range $[-s/2, s/2]$, where $s$ is the scale factor. Due to the round-to-nearest behavior, the error has zero mean and RMS $s^2/12$, which represents the noise power. Thus, the SNR for per-tensor quantization is given by:

$$\text{SNR}_{\text{tensor}} := 10 \log(\frac{12\sigma_X^2}{s^2}) = 10 \log(\frac{12\sigma_X^2 (\Delta_{\max}^{\text{FP8}})^2}{\max(|\boldsymbol{X}|)^2}). \tag{5}$$

Next, consider per-group quantization with $N$ groups, each with a group size of 128. Each group $g$ is quantized independently using its own scaling factor $s_g$. Since the quantization error within each group is assumed to be uniformly distributed, averaging the noise across all groups yields $P_{\text{noise}} = \frac{1}{12N} \sum_g s_g^2$. Hence, the SNR for per-group quantization is given by:

$$\text{SNR}_{\text{group}} := 10 \log(\frac{12N\sigma_X^2}{\sum_g s_g^2}) = 10 \log(\frac{12N\sigma_X^2 (\Delta_{\max}^{\text{FP8}})^2}{\sum_g \max(|\boldsymbol{X}_g|)^2}). \tag{6}$$

This formulation shows that finer granularity (larger $N$) can yield a higher SNR by better adapting to local signal variations. Through comparing Equation (5) and Equation (6), we observe that $\text{SNR}_{\text{per-group}} > \text{SNR}_{\text{per-tensor}}$ since $\sum_g \max(|\boldsymbol{X}_g|)^2/N \le \max(|\boldsymbol{X}|)^2$, indicating that per-group quantization offers higher quantization fidelity than per-tensor quantization due to its finer scaling adaptation. Now, consider the two-level microscaling quantization scheme used in MOSS. In this design, the effective scaling factor for a subgroup is given by $s_{\text{MX}} = s_{\text{global}} \cdot ss_i$, where $s_{\text{global}}$ is the global per-tensor scale, and $ss_i$ is the second-level microscale for each group, defined in Equation (3). Assuming the input tensor is divided into $N_g$ micro groups, each of size 32, the resulting quantization noise power becomes $P_{\text{noise}} = \frac{1}{12N_g} \sum_g s_{\text{global}}^2 \cdot ss_i^2$. Thus, the corresponding SNR for MOSS's two-level microscaling is:

$$\text{SNR}_{\text{MOSS}} := 10 \log(\frac{12N_g\sigma_X^2}{\sum_i s_{\text{global}}^2 \cdot ss_i^2}) \simeq 10 \log(\frac{12N_g\sigma_X^2 (\Delta_{\max}^{\text{FP8}})^2}{\sum_i \max(|\boldsymbol{X}_i|)^2}), \tag{7}$$

where $N_g = 4N$ with micro group size 32. Since each subgroup in MOSS contains fewer values, its maximum magnitude is upper bounded by that of the per-group quantization, i.e., $\max(|\boldsymbol{X}_i|) \le \max(|\boldsymbol{X}_g|)$. Consequently, by averaging over all subgroups, we have $\frac{\sum_{i=1}^{N_g} \max(|\boldsymbol{X}_i|)^2}{N_g} < \frac{\sum_{g=1}^{N} \max(|\boldsymbol{X}_g|)^2}{N}$, which implies $\text{SNR}_{\text{MOSS}} > \text{SNR}_{\text{per-group}}$. Therefore, combining with the SNR formulations for each scheme, we arrive at: $\text{SNR}_{\text{per-tensor}} < \text{SNR}_{\text{per-group}} < \text{SNR}_{\text{MOSS}}$. This inequality shows that the two-level microscaling in MOSS achieves lower quantization error than both per-tensor and per-group quantization. This formulation demonstrates that MOSS retains high quantization fidelity by combining coarse-grained global scaling with lightweight power-of-two microscales, effectively balancing accuracy and hardware efficiency.

## 3.2 Automatic Scaling

Various scaling techniques have been proposed to adapt tensors to the limited dynamic range of the FP8 format (Nvidia, 2025; Blake et al., 2023). However, these methods often introduce training overhead due to additional memory accesses and max-reduction operations. For example, just-in-time scaling requires reading all FP32 values from High Bandwidth Memory (HBM) to compute the maximum absolute value for quantization. The resulting FP8 values are then written back to HBM, only to be read again during subsequent MMAs, incurring significant latency and bandwidth costs.

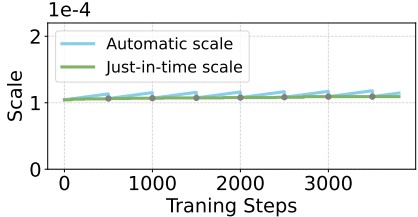

Figure 4: Automatic scaling trend under interval = 500.

Table 1: Time usage to calculate per-tensor scaling factors for parameters. Automatic scaling in MOSS incurs negligible and constant runtime overhead compared to just-in-time scaling.

| Tensor Size | Just-in-time Scaling | Automatic Scaling |
|---|---|---|
| $11008 \times 16384$ | 0.54ms | 0.02ms |
| $11008 \times 8192$ | 0.32ms | 0.02ms |
| $4096 \times 12288$ | 0.17ms | 0.02ms |
| $4096 \times 4096$ | 0.08ms | 0.02ms |

To address this inefficiency, we propose a novel automatic scaling approach that estimates the evolution of scaling factors for parameters throughout training. LLMs are typically trained using Adam or its variants (Kingma & Ba, 2014; Loshchilov & Hutter, 2017), which maintain additional copies of model weights, gradients, and first- and second-order momentums for parameter updates. A key advantage of these optimizers is their ability to produce stable updates, with changes to model parameters bounded by the step size $\eta$, regardless of gradient volatility. By exploiting this bounded-update property, we propose a novel automatic scaling scheme that can predict and adjust scaling factors ahead of time, eliminating the need for repeated memory reads and writes, and thus avoiding runtime overhead.

**Theorem 2.** During training, the effective update at time stamp $t$ has two upper bounds:

$$|\Delta_t| \leq \begin{cases} \eta \cdot \frac{1-\beta_1^t}{\sqrt{1-\beta_2^t}}, & \text{if } 1 - \beta_1^t > \sqrt{1 - \beta_2^t} \\ \eta, & \text{otherwise,} \end{cases} \tag{8}$$

where $\beta_1^t$ and $\beta_2^t$ denote $\beta_1$ and $\beta_2$ raised to the power of $t$, respectively.

**Proof 2.** Based on the weight update rule in Equation (1), the effective parameter update at timestep $t$ is given by $\Delta_t = \eta \cdot \hat{\mathbf{m}}_t / \sqrt{\hat{\mathbf{v}}_t}$, assuming $\epsilon \to 0$. This expression reveals that $|\Delta_t|$ is governed by three key components:

$$|\Delta_t| = \eta \cdot (|\frac{\mathbf{m}_t}{\sqrt{\mathbf{v}_t}}|) \cdot (\frac{1 - \beta_1^t}{\sqrt{1 - \beta_2^t}}). \tag{9}$$

To analyze the bound of $|\Delta_t|$, we first examine the term $|\mathbf{m}_t / \sqrt{\mathbf{v}_t}|$. During LLM training, both the first-order moment $\mathbf{m}_t$ and the second-order moment $\mathbf{v}_t$ are initialized to zero, i.e., $\mathbf{m}_0 = \mathbf{0}$ and $\mathbf{v}_0 = \mathbf{0}$. As training progresses, $\mathbf{m}_t$ becomes the exponential moving average (EMA) of the gradients, i.e., $\mathbf{m}_t = \mathbb{E}[\mathbf{g}]_{\text{EMA}}$, and $\mathbf{v}_t$ becomes the EMA of the squared gradients, i.e., $\mathbf{v}_t = \mathbb{E}[\mathbf{g}^2]_{\text{EMA}}$. Applying Jensen's inequality (Abramovich et al., 2004) to the convex function $f(x) = x^2$ yields the relation $(\mathbf{m}_t)^2 \leq \mathbf{v}_t$, which in turn implies that $|\mathbf{m}_t / \sqrt{\mathbf{v}_t}| \leq 1$.

Next, consider the term $(1 - \beta_1^t) / \sqrt{1 - \beta_2^t}$ in Equation (8). The first case of Equation (8) corresponds to an extreme scenario of gradient sparsity: when the gradient is zero at all previous timesteps and nonzero only at the current step, which is extremely unusual in practice. Under this worst-case condition, the effective update is bounded by $|\Delta_t| \leq \eta \cdot (1 - \beta_1^t) / \sqrt{1 - \beta_2^t}$. In less sparse scenarios where gradients appear across multiple steps, the EMA terms accumulate over time, leading to smaller normalized updates. As a result, the effective update $|\Delta_t|$ is generally bounded by $\eta$.

In more typical cases, the effective update $|\Delta_t|$ is strictly smaller than the step size $\eta$. For example, when $1 - \beta_1^t = \sqrt{1 - \beta_2^t}$, and given that $|\hat{\mathbf{m}}_t / \sqrt{\hat{\mathbf{v}}_t}| \leq 1$, it follows directly that $|\Delta_t| < \eta$. In practice, during LLM training, it is common to have $1 - \beta_1^t < \sqrt{1 - \beta_2^t}$ because LLMs typically adopt smaller values for $\beta_2$ (e.g., $\beta_2 = 0.95$) (Grattafiori et al., 2024; Groeneveld et al., 2024). This choice keeps the second-order moment estimate more responsive to recent gradients, helping to prevent both exploding updates and excessively slow learning. Overall, the effective parameter update at each timestep is approximately bounded by the step size $\eta$, i.e., $|\Delta_t| \leq \eta$. This behavior can be interpreted as implicitly defining a trust region around the current parameter values, within which the optimizer assumes the gradient estimate remains informative and reliable.

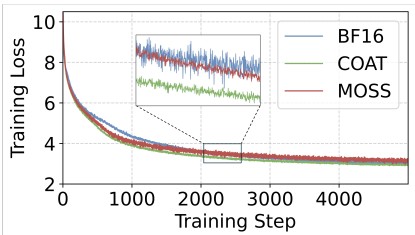

Figure 5: OLMo-7B pretraining curve.

Table 2: OLMo-7B pretraining performance on downstream tasks. For system performance, we report results of training throughput. For model performance, we present the Perplexity on WikiText-103, C4, and Pile.

| Models | Throughput (tokens/s) | Model Performance (PPL) | | |
|---|---|---|---|---|
| | | WikiText-103 | C4 | Pile |
| BF16 | 33,805 | 39.59 | 30.59 | 25.18 |
| COAT | 40,416(+19.6%) | 40.62 | 30.89 | 26.05 |
| MOSS | 45,374(+34.2%) | 40.96 | 30.63 | 25.08 |

Since the step size $\eta$ is typically known in advance, we can infer the potential evolution of scaling factors throughout training. At initialization, the scaling factor is determined by the maximum absolute value of the model parameters. Based on the update bound in Equation (8), the maximum absolute value of the weights at timestep $t$ satisfies: $\max(|\boldsymbol{W}_t|) \leq \max(|\boldsymbol{W}_0|) + \eta \cdot t$. This relation provides a predictable upper bound on parameter magnitude over time. Consequently, the scaling factor for model weights at timestep $t$ can be determined as

$$s_t = \frac{\max(|\boldsymbol{W}_0|) + \eta \cdot t}{\Delta_{\max}^{\mathrm{FP8}}} = s_0 + \frac{\eta \cdot t}{\Delta_{\max}^{\mathrm{FP8}}}, \tag{10}$$

where $s_t$ denotes the scaling factor for the weight block at timestep $t$, and $s_0$ is the initial scaling factor, determined via a max-reduction operation at initialization. This formulation eliminates the need for dynamic max-reduction at every step, while still ensuring that the scaled weights remain within the representable range of FP8. As shown in Table 1, our proposed automatic scaling introduces negligible overhead, operating in constant time across all tensor sizes. In contrast, just-in-time scaling incurs significant computational costs due to its max reduction process, especially for large tensors.

To maintain training stability and numerical accuracy, MOSS performs dynamic re-scaling at fixed intervals. Specifically, within each interval, scaling factors are updated using the rule defined in Equation (10), incurring negligible memory access overhead. At the end of each interval, MOSS rescales the current weight tensors and updates the associated per-tensor scaling factors accordingly. The effectiveness of this approach is illustrated in Figure 4, where an interval length of 500 is used. We observe that the scaling trajectory produced by automatic scaling consistently lies above that of just-in-time scaling, helping to ensure that values remain well within the representable FP8 range and reducing the risk of overflows or underflows. Notably, the two scale curves remain relatively close throughout training, indicating that MOSS maintains a high degree of precision even with infrequent updates. The choice of interval length presents a trade-off between scale estimation precision and computational efficiency. While the above theoretical analysis assumes the use of the Adam optimizer, the same principles apply to AdamW. Additional discussion can be found in the Appendix. This periodic scaling strategy enables MOSS to perform automatic parameter quantization with minimal overhead, offering a practical alternative to more expensive just-in-time or delayed scaling methods.

## 4 Experiments

### 4.1 Experimental Setup

Setups. We train the language models using the AdamW optimizer (Loshchilov & Hutter, 2017), an extension of Adam (Kingma & Ba, 2014) with decoupled weight decay. Following common practice, we use decay rates of $\beta_1 = 0.9$, $\beta_2 = 0.95$, and a weight decay of 0.1. The learning rate, beginning from $2e^{-4}$, follows a cosine decay schedule, with the final learning rate set to 10% of the peak value. Models are trained on a total of 22B tokens with a global batch size of 256, which is sufficient to preserve the Gaussian distribution of weight updates. The input sequence length is fixed at 2048 tokens, and a warm-up phase of 2,000 iterations is applied. Additional model configurations and training details are provided in the Appendix. Training is conducted on 8 NVIDIA H800 GPUs, which do not natively support the microscaling data format available on the newer Blackwell GPU series. To enable MXFP8 computation on standard GPUs such as NVIDIA Hopper, we implement custom matrix multiplication kernels using Triton (OpenAI, 2025). We use the

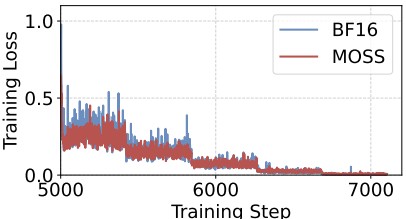

Figure 6: LLaMA-2-7B fine-tuning.

Table 3: LLaMA-2-7B fine-tuning performance and evaluation accuracy on downstream tasks. System performance is assessed with model throughput, while model performance is evaluated on Mathematics, GSM8K, NumGLE using accuracy.

| Models | Throughput (samples/s) | Model Performance (Acc%) | | |
|--------|------------------------|-------------|--------|---------|
| | | Mathematics | GSM8K | NumGLUE |
| BF16 | 168.2 | 52.3 | 65.2 | 58.7 |
| MOSS | 241.8(+43.8%) | 52.8 | 64.7 | 59.4 |

proposed two-level microscaling method to quantize activations and per-tensor quantization with automatic scaling for model weights in linear layers.

Baselines and Datasets. We compare MOSS against BF16 training and the COAT baseline (Xi et al., 2024) to evaluate its effectiveness in both LLM pretraining and fine-tuning tasks. Transformer Engine (Nvidia, 2025) also adopts MXFP8 for quantization; however, its microscaling strategy is supported only on NVIDIA Blackwell GPUs, which limits its broader applicability. For LLM pretraining, we train the OLMo-7B model on the Dolma corpus (Soldaini et al., 2024) and report perplexity on standard benchmarks including WikiText-103 (Merity et al., 2016), C4 (Raffel et al., 2020), and The Pile (Gao et al., 2020). For LLM fine-tuning, we focus on mathematical reasoning tasks by fine-tuning a LLaMA-2-7B model (Grattafiori et al., 2024) on the MAmmoTH dataset (Yue et al., 2023). We then evaluate the instruction-tuned model on three downstream benchmarks: Mathematics (Davies et al., 2021), GSM8K (Cobbe et al., 2021), and NumGLUE (Mishra et al., 2022).

## 4.2   LLM Pre-Training

We begin by comparing the performance of OLMo-7B models (Groeneveld et al., 2024) pre-trained using FP8 mixed precision with those trained using BF16. The pre-training loss over training steps is shown in Figure 5. All models use identical training configurations and hyperparameters as specified in the official report. As illustrated in Figure 5, the loss curves of BF16, COAT, and our proposed method, MOSS, closely align, exhibiting nearly identical trends. These results clearly demonstrate that the proposed FP8 mixed-precision scheme can match the training performance of the widely adopted higher-precision BF16 baseline.

We further evaluate the pre-trained models on a diverse set of downstream tasks, with results summarized in Table 2. The proposed MOSS-trained models achieve zero-shot performance comparable to their BF16 counterparts, demonstrating that MOSS preserves both model accuracy and in-context learning capabilities at a level on par with high-precision training. In terms of efficiency, Table 2 also reports training throughput, where MOSS delivers a 34% speedup over BF16 and a 12% improvement over COAT, underscoring its superior performance. To assess convergence behavior, we extend pre-training of the OLMo-7B model with MOSS for an additional 30,000 steps (totaling 35,000 steps), presented in the Appendix, to validate the robustness and scalability of the MOSS framework.

## 4.3   LLM Fine-Tuning

We further evaluate the proposed FP8 low-precision training scheme in the context of instruction tuning for LLMs. To ensure a fair comparison, we adopt the same fine-tuning setup as ToolBench (Guo et al., 2024), using the open-source LLaMA-2-7B (Grattafiori et al., 2024) as the base model. Experiments are conducted on the math-focused MAmmoTH dataset (Yue et al., 2023), fine-tuned for 5 epochs. During fine-tuning, MOSS matches the performance of the BF16 baseline, with loss curves that are either closely aligned or exhibit slightly improved stability (Figure 6). Downstream task accuracies are reported in Table 3. These results further demonstrate the numerical precision and robustness of MOSS, confirming that the proposed FP8 quantization scheme generalizes well beyond pre-training and remains effective in fine-tuning settings.

To further evaluate the scalability of MOSS on larger models, we fine-tune Qwen-3-14B and Qwen-3-32B under both BF16 and proposed MOSS framework, and report their downstream reasoning performance. Experiments are conducted on curated dataset of MegaScience Fan et al. (2025) and NuminaMath-CoT Li et al. (2024). All metrics are averaged over three

Table 4: Downstream evaluation accuracy (%) of Qwen-3-14B and Qwen-3-32B fine-tuned under BF16 and MOSS. All results are averaged over three independent runs.

| Model | Precision | MATH500 | GPQA-Diamond | C-Eval | AIME24 | MMLU-Redux |
|-------|-----------|---------|--------------|--------|--------|------------|
| Qwen-3-14B | BF16 | 96.8 | 62.1 | 86.2 | 79.1 | 88.6 |
| | MOSS | 96.7 | 61.5 | 86.7 | 79.4 | 88.5 |
| Qwen-3-32B | BF16 | 98.3 | 64.2 | 88.2 | 79.3 | 89.6 |
| | MOSS | 98.1 | 64.4 | 87.8 | 78.8 | 88.6 |

independent fine-tuning runs to ensure statistical reliability. Table 4 presents the accuracy on five representative reasoning and knowledge benchmarks. Across both model sizes, MOSS matches the BF16 baseline on every task, demonstrating negligible performance loss despite operating at reduced precision. Notably, MOSS maintains strong stability on long-horizon reasoning tasks (e.g., MATH500, AIME24), where quantization methods often struggle with scale drift or cumulative error. These results indicate that MOSS scales effectively to 30B-class models while preserving competitive downstream performance.

## 4.4 Memory and Communication Gains

We conduct detailed profiling of peak activation memory usage and inter-GPU communication volume during LLaMA-2-7B fine-tuning on an 8×H200 cluster (3.2 TB/s aggregate NVLink bandwidth). All experiments use a per-GPU batch size of 4, a sequence length of 4096, and FSDP with ZeRO-2. Communication statistics are collected using the NCCL profiler and averaged over 1,000 training steps.

The measurements in Table 5 highlight the substantial memory-efficiency benefits introduced by MOSS. Even on bandwidth-rich H200 GPUs, MOSS delivers a 1.8× reduction in peak activation memory, confirming that the improvement arises from its FP8 microscaling algorithm rather than from hardware-level memory bandwidth limitations. We additionally evaluate compute–communication overlap using the PyTorch Profiler. MOSS achieves an overlap ratio exceeding 83%, indicating that it preserves strong scalability under high-speed NVLink interconnects and effectively reduces inter-GPU communication overhead during distributed training.

Table 5: Memory footprint and communication efficiency across BF16, COAT, and MOSS.

| Models | Memory | | | Communication | |
|--------|--------|--|--|---------------|--|
| | Peak Activation (GB) | AllReduce Volume (GB/step) | Saving | AllReduce Latency (ms) | Overlap Ratio (%) |
| BF16 | 42.3 | 3.84 | 1.00× | 24.8 | 71.3 |
| COAT | 28.6 | 3.12 | 1.48× | 18.6 | 78.5 |
| MOSS | 23.5 | 2.74 | 1.80× | 16.2 | 83.4 |

## 4.5 Ablation Study

### 4.5.1 GEMM Performance

To evaluate the efficiency of the GEMM kernel in MOSS, we conduct an ablation study comparing our implementation with several state-of-the-art FP8 quantization schemes, including Transformer Engine (TE)(Nvidia, 2025), DeepGEMM from DeepSeek(Zhao et al., 2025), and COAT (Xi et al., 2024). This evaluation focuses on both raw GEMM performance and the practical trade-offs in quantization and hardware deployment.

Table 6: Runtime of quantized FP8 GEMM on NVIDIA H800 GPUs.

| M | N | K | Runtime(ms) | | | |
|---|---|---|----|------|----------|------|
| | | | TE | COAT | DeepSeek | MOSS |
| 2048 | 7168 | 4096 | 0.26 | 0.32 | 0.2 | 0.25 |
| 2048 | 7168 | 11008 | 0.63 | 0.86 | 0.46 | 0.67 |
| 4096 | 2048 | 7168 | 0.37 | 2.27 | 0.23 | 0.45 |
| 4096 | 4096 | 8192 | 0.52 | 2.63 | 0.39 | 0.60 |
| 4096 | 4096 | 12288 | 0.75 | 3.91 | 0.62 | 0.72 |
| 5120 | 5120 | 10240 | 1.16 | 5.57 | 0.68 | 1.04 |
| 8192 | 8192 | 8192 | 2.16 | 10.54 | 1.23 | 1.98 |
| Avg | | | 0.84 | 3.73 | 0.54 | 0.77 |

As shown in Table 6, MOSS achieves GEMM performance that is comparable to TE. Importantly, MOSS significantly outperforms COAT in GEMM throughput, highlighting the efficiency of our low-overhead quantization strategy. In comparison to DeepGEMM, our GEMM kernel runtime is relatively higher. This is expected, as DeepSeek leverages the Increasing Accumulation Precision technique to reduce dequantization overhead and employs

extensive hardware-specific optimizations targeting NVIDIA's Hopper GPUs. In contrast, our implementation is not tailored to Hopper GPUs, which lack the native Tensor Core support for MX data format. Despite this, MOSS remains competitive in performance across widely available hardware and maintains robust generalizability without relying on hardware-specific enhancements.

### 4.5.2 Quantization Fidelity

To evaluate the fidelity of different quantization strategies, we extracted activation tensors from LLaMA-2-7B during the actual fine-tuning process. Specifically, we sampled tensors from three critical layers—attention outputs, FFN intermediate activations, and LayerNorm inputs—every 100 training steps over a total of 10,000 steps. For each sample, we computed the signal-to-noise ratio (SNR) under three quantization schemes: 1) per-tensor quantization; 2) per-group quantization (group size = 128); 3) two-level microscaling (MOSS, micro group size = 32). The results, segmented into early (step < 2K) and late (step > 8K) training stages, are summarized in Table 7.

Table 7: SNR (dB) of activation tensors across different layers and quantization strategies during LLaMA-2-7B fine-tuning.

| Layer | Per-Tensor | | Per-Group | | MOSS (Two-Level) | |
|---|---|---|---|---|---|---|
| | Early | Late | Early | Late | Early | Late |
| Attention Output | 28.4 | 26.7 | 34.8 | 33.2 | 37.6 | 36.1 |
| FFN Intermediate | 25.9 | 24.1 | 32.3 | 30.7 | 35.9 | 35.3 |
| LayerNorm Input | 31.2 | 29.5 | 36.7 | 35.1 | 39.4 | 38.0 |
| Geometric Mean | 28.3 | 26.6 | 34.5 | 32.9 | 37.5 | 36.0 |

The experimental results confirm our theoretical analysis in Section 3.1. Specifically, MOSS consistently achieves superior quantization fidelity, improving SNR by 3.0–3.4 dB over per-group quantization and 9.2–9.4 dB over per-tensor quantization. More over, this SNR advantage is maintained across all layer types and throughout the training process, demonstrating the robustness of our two-level microscaling approach. These findings provide strong empirical support for the effectiveness of MOSS and validate the conclusions discussed in Section 3.1.

## 5 Related Works

### 5.1 Low-Precision Training

Low-precision training refers to using low-precision bits, such as FP16/BF16, in the training process, as opposed to using FP32 full-precision throughout the training process. It is one of the most prominent directions in modern deep learning, as it offers reductions in computation costs, memory footprint and communication overhead (Micikevicius et al., 2017). Most existing training frameworks, e.g., Megatron-LLM (Shoeybi et al., 2019) and Deep-Speed (Rasley et al., 2020), adopt BF16 in LLM training by default. With the release of Nvidia Hopper GPUs, FP8 training (Micikevicius et al., 2022) becomes the next-generation low-precision technique. Transformer Engine (Nvidia, 2025) from NVIDIA is the first framework designed for FP8 mixed-precision training, however, it only supports FP8 compute for linear layers while leaving all other operations in full precision. FP8-LM (Peng et al., 2023) extends FP8 quantization to gradients and first-order optimizer states, further improving the training throughput. However, they fail to reduce the memory footprint of activations, not fully exploiting the potential of memory and communication benefits of FP8 in fully sharded distributed training scenarios. COAT (Xi et al., 2024) and DeepSeek-V3 (Liu et al., 2024) both extend the per-group FP8 quantization to activations, achieving mixed granularity quantization. However, the fine-grained quantization strategy incurs additional computation cost in the dequantization process.

### 5.2 Scaling Techniques

Quantization transforms high-precision values within a given range into lower-precision values of a different range by tensor scaling. Scaling factor determination during training

typically follows one of two strategies: Delayed scaling estimates the current scaling factor using a historical record of maximum absolute values from previous iterations. This approach assumes statistical consistency across iterations, making it vulnerable to outliers that violate this assumption and potentially destabilize training (Fishman et al., 2024). In contrast, just-in-time scaling dynamically adjusts scaling factors based on the current data distribution, typically using on-the-fly max-reduction. While more adaptive, this method introduces significant computational overhead—particularly in FP8 training—where multiple memory accesses can offset the performance gains of low-precision computation. Beyond these non-automatic scaling approaches, Unit Scaling (Blake et al., 2023) recently introduced an automatic scaling method that maintains unit variance in weights and activations by applying static scaling constants throughout the network. However, Unit Scaling requires manual modification of the computational graph to insert unit-scaled operations and preserves certain "critical matmuls" (e.g., attention output projection and FFN down projection) in BF16 for stability. In contrast, MOSS offers a generic and automated scaling mechanism that integrates seamlessly into modern LLMs, achieving full FP8 precision without requiring precision fallbacks or manual graph modifications.

## 6  Conclusion

In this work, we present MOSS, a novel framework for efficient and accurate FP8 training of large language models. To address the significant dequantization overhead on slower CUDA cores within the GEMM main loop, we introduce a two-level microscaling quantization scheme that minimizes overhead while preserving high numerical precision at fine granularity. Then, by leveraging the observation that parameter updates are bounded by the learning rate, we propose an automatic scaling mechanism that eliminates the runtime cost of online scaling. Through extensive experiments on a 7B parameter model, MOSS achieves training accuracy on par with BF16 baselines while delivering a 34% improvement in throughput. Furthermore, the framework generalizes effectively to downstream fine-tuning tasks, demonstrating its versatility beyond pre-training. Overall, MOSS offers a practical and hardware-friendly solution for scalable FP8 training. Future work could further explore combining our proposed MOSS with other low-precision compression methods to reduce computation overhead in non-linear layers.

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

## A    Model Configuration

For all experiments, we adopt the default hyperparameters in the official training recipe. Table 8 presents the details of model configurations.

Table 8: Model sizes, architectures, and training hyperparameters.

| Model | Params | Dimension | #heads | #layers | Learning rate | Weight decay | Sequence length | Batch size |
|---|---|---|---|---|---|---|---|---|
| OLMo | 7B | 4096 | 32 | 32 | 3e-4 | 0.1 | 2048 | 2 |
| LLaMA | 7B | 4096 | 32 | 32 | 5e-5 | 0 | 4096 | 2 |

## B    MOSS Pretraining Performance

We evaluate the training performance of MOSS by training it for 36,000 steps on approximately 144B tokens. Due to resource constraints, we do not perform full pre-training for the BF16 and COAT baselines. As shown in Figure 7, the training curve demonstrates the stability of our proposed MOSS framework.

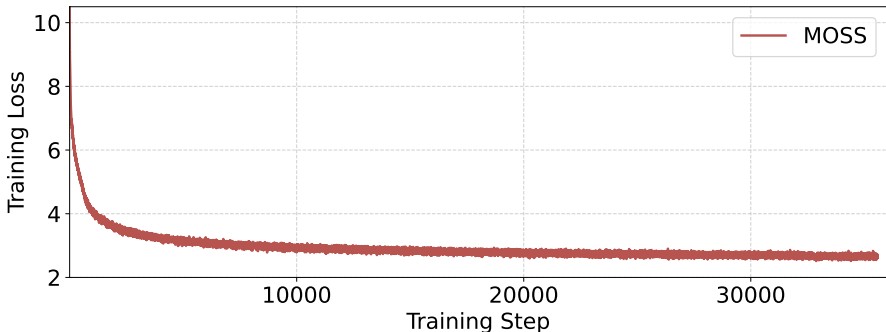

Figure 7: Loss Curve of MOSS FP8 Training.

## C    Discussion of Bound in AdamW Optimizer

Compared to Adam optimizer (Kingma & Ba, 2014), AdamW (Loshchilov & Hutter, 2017) decouples the $L_2$ regularization from the loss function and directly inject the weight decay, i.e., $-\eta\lambda\boldsymbol{W}_t$, into the weight update. In this way, AdamW enforces stable shrinkage and tends to converge to smaller absolute values, particularly in high-magnitude regimes. Based on the update rule in Equation (1), we have

$$\mathbf{W}_{t+1} = (1 - \eta\lambda)\mathbf{W}_t - \eta\frac{\hat{\mathbf{m}}_t}{\sqrt{\hat{\mathbf{v}}_t} + \epsilon} < \mathbf{W}_t - \eta\frac{\hat{\mathbf{m}}_t}{\sqrt{\hat{\mathbf{v}}_t} + \epsilon}. \tag{11}$$

Therefore, the maximum absolute value of the weights at timestep $t$ remains bounded by $\max(|\boldsymbol{W}_t|) \leq \max(|\boldsymbol{W}_0|) + \eta \cdot t$, allowing the scaling factor update rule in Equation (10) to be directly applied to AdamW-optimized training.

## D    Ablation Study on different intervals

To understand how the update interval affects the trade-off between computational overhead and model accuracy, we conduct an ablation study on LLaMA-2-7B fine-tuning with a range of interval values. Table 9 reports the corresponding scaling overhead, effective throughput, and downstream accuracy on NumGLUE.

Our findings reveal a clear pattern. Extremely frequent updates—such as the just-in-time configuration that recomputes scaling every step—introduce substantial overhead (3.8 ms/step) without improving throughput. In contrast, moderate update intervals in the range of 100–500 steps reduce scaling overhead to a negligible level (<0.03 ms/step),

Table 9: Impact of scaling interval on overhead, throughput, and accuracy during fine-tuning.

| Method | Interval | Scaling Overhead (ms/step) | Effective Throughput | Accuracy (NumGLUE) |
|--------|----------|---------------------------|---------------------|--------------------|
| BF16 | – | 0 | 1.0× | 58.7% |
| JIT | 1 | 3.8 | 1.0× | 59.3% |
| MOSS | 100 | 0.03 | 1.043× | 59.4% |
| MOSS | 500 (default) | 0.02 | 1.044× | 59.4% |
| MOSS | 2000 | 0.01 | 1.044× | 58.1% |

yielding the highest effective throughput (1.043–1.044×) while matching or slightly exceeding BF16 accuracy. When the interval becomes too large (e.g., 2000), we observe a mild drop in accuracy due to insufficient tracking of scale drift, although the throughput remains comparable.

Overall, these results demonstrate that MOSS is highly robust to the choice of update interval across a broad regime. The default value of 500 steps strikes an optimal balance, minimizing scaling overhead while preserving accuracy and maximizing training efficiency.

## E  End-to-End Throughput Improvement of Automatic Scaling

To quantify the end-to-end throughput benefits of different weight quantization strategies, we compared three scaling methods during LLaMA-2-7B fine-tuning on 8×H800 GPUs:

1. Just-in-time (JIT) scaling: Performs on-the-fly max-reduction at every forward pass.

2. Delayed scaling: Uses historical maximum values with a moving window.

3. Automatic scaling (MOSS): Predicts scaling factors based on the learning rate schedule (our method).

Table 10: End-to-end throughput of different weight scaling strategies during LLaMA-2-7B fine-tuning on 8×H800 GPUs.

| Method | Scaling Overhead (ms) | Total Step Time (ms) | Throughput (token/s) | Speedup |
|--------|----------------------|---------------------|---------------------|---------|
| JIT scaling | 3.8 | 68.5 | 38,642 | 1.0× |
| Delayed scaling | 1.2 | 65.9 | 40,182 | 1.04× |
| MOSS (Automatic) | 0.2 | 63.1 | 41,998 | 1.087× |

Table 10 summarizes the end-to-end throughput during fine-tuning (batch size = 4, sequence length = 4096), averaged over 1,000 steps. As shown in Table 10, our automatic scaling method achieves an 8.7% end-to-end throughput improvement over the JIT baseline. This improvement primarily arises from eliminating the costly memory access pattern in JIT scaling, which requires reading the entire FP32 weight tensor from HBM for max-reduction. In contrast, MOSS incurs only negligible, constant-time overhead while maintaining accuracy.

Furthermore, to demonstrate the scalability and accuracy, we evaluated the downstream task accuracy of our automatic scaling method against just-in-time scaling under identical fine-tuning settings (LLaMA-2-7B on the MAmmoTH dataset). The results are summarized in Table 11.

Table 11: Downstream task accuracy comparison between JIT scaling and MOSS (Automatic). Differences are reported relative to JIT scaling.

| Scaling Method | Mathematics | GSM8K | NumGLUE |
|----------------|-------------|-------|---------|
| JIT Scaling | 52.6% | 64.9% | 59.2% |
| MOSS (Automatic) | 52.8% | 64.7% | 59.4% |
| Difference | +0.2% | -0.2% | +0.2% |

The observed differences in accuracy are within the statistical noise margin (±0.3% across three independent seeds), confirming that the theoretical upper bound in Equation (10)

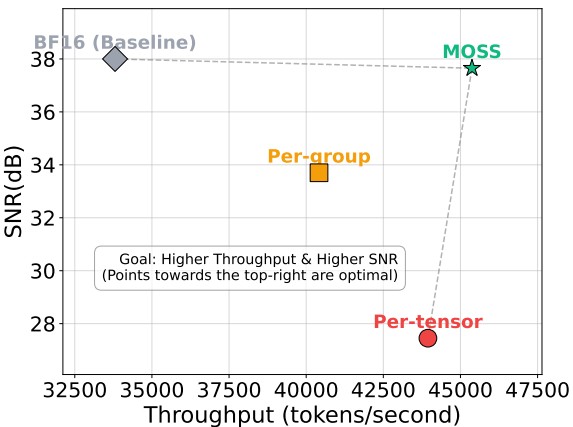

Figure 8: Quantization Technique Trade-offs: Throughput vs. Precision.

and the dynamic re-scaling mechanism provide sufficient numerical precision for practical training. Consequently, the 8.7% end-to-end throughput improvement achieved by automatic scaling is obtained without any degradation in model quality.

## F  Visualization of Quantization Methods

To give an overview about the trade-off between quantization fidelty and throughput, we visualize the performance landscape of different quantization strategies on a 2D plane (as shown in Figure 8), where the X-axis represents throughput (tokens/second) and the Y-axis represents quantization fidelity (SNR in dB). This visualization integrates our ablation study results and clearly illustrates that MOSS lies on the desirable Pareto frontier, effectively avoiding the core limitations exhibited by prior approaches.

## G  Limitations

While MOSS demonstrates promising results in FP8 training efficiency, our work has several limitations. First, our focus on GEMM operations leaves room for improvement in non-linear layers (e.g., activations, LayerNorm), where low-precision overheads persist. Then, the current implementation optimizes for CUDA cores, and its performance gains may not generalize fully to other architectures, i.e., TPUs or AMD GPUs, without further adaptation. Finally, although MOSS achieves parity with BF16 on a 7B model, its efficacy at larger scales (e.g., 100B+ parameters) remains to be validated, particularly where gradient dynamics may differ.

## H  Broader impact

Our proposed algorithm, MOSS, can improve efficiency and reduce the energy consumption of training neural networks, which helps reduce the carbon footprint caused by deep learning. By reducing computational costs, MOSS could lower barriers to training and fine-tuning large models, benefiting resource-constrained researchers. Moreover, widespread adoption of FP8 training might necessitate hardware/software co-design, incentivizing vendors to prioritize FP8 support and standardization.

