# OpenReview forum: "MOSS: Efficient and Accurate FP8 LLM Training with Microscaling and Automatic Scaling"
_ICLR.cc/2026/Conference — ICLR 2026 Poster_

### Official Review · Reviewer_ZzQN · 2025-10-29

**Soundness:** 2
**Presentation:** 4
**Contribution:** 3
**Rating:** 6
**Confidence:** 5

**Summary:**

This paper introduces MOSS, a specialized strategy designed to accelerate FP8 training for Large Language Models (LLMs). Traditional training frameworks often face a trade-off between quantization accuracy and computational overhead, dictated by the granularity of quantization. MOSS innovatively proposes methods to better balance these two factors within the FP8 training framework: First, for sensitive activation tensors, it employs a two-step quantization strategy using coarse-grained FP32 quantization scale factors combined with fine-grained MXFP quantization factors, addressing the high computational overhead associated with the dequantization phase in traditional per-group quantization. Second, for weight tensors, it adopts a strategy of real-time prediction of learning rates during training, thereby avoiding the computationally expensive real-time calculation of scaling factors. Experiments demonstrate that their strategy achieves a throughput improvement of approximately 35% over a BF16 baseline method while maintaining the training performance of a 7B model.

**Strengths:**

1.  **Important Problem, Insightful Perspective:** The trade-off between quantization accuracy and computational overhead is a central focus in the field of quantized training. The paper's focus on FP8 training is one of the most important and active problems in this area, and the proposed solution for balancing these factors is highly instructive.
2.  **Sufficient Novelty:** The two-step quantization framework using coarse-grained FP32 scale factors combined with fine-grained MXFP scale factors effectively addresses the significant computational overhead associated with the dequantization phase within traditional per-group frameworks on CUDA Cores. This approach cleverly leverages high-performance Tensor Cores. The framework for automatic weight quantization is novel; by integrating with the learning rate within the optimizer, it directly bypasses the step of calculating scaling factors in real-time, offering potential for acceleration.
3.  **Relatively Complete Theoretical Analysis:** The theoretical analysis is convincing, whether discussing the motivation for the two-step quantization or the mathematical proof regarding the combination of autoscaling and learning rate. These analyses also incorporate practical considerations. For example, periodically recalculating the true scaling factor during autoscaling maintains mathematical correctness while adhering to the practical requirements of quantized training.

**Weaknesses:**

The paper has few shortcomings elsewhere, but the ablation experiments regarding the critical aspect of "computational efficiency" are insufficient. For example:

1.  **End-to-End Throughput Impact of Automatic Weight Scaling:** How much throughput improvement does automatic scaling on weights *actually* provide in an end-to-end scenario? The paper only shows improvements for GeMM alone (Table 1). While automatic scaling is certainly better than in-time scaling for GeMM separately, in-time scaling might not cause significant throughput reduction end-to-end. If the accuracy degradation caused by potentially less precise weight quantization under automatic scaling outweighs the throughput benefits end-to-end, the approach would be counterproductive. The authors are encouraged to provide analysis on this aspect.
2.  **Two-Stage Scaling Strategy for Activations:** Although the authors theoretically prove that MOSS can yield higher Signal-to-Noise Ratio (SNR), experimental validation is lacking. Ideally, a training curve like Figure 5 would be helpful, though I fully understand computational resource constraints. Alternatively, extracting activation tensors during the actual training process and providing quantized SNR results for per-group, per-tensor, and MOSS would better support the conclusions in Section 3.1.
3.  **Overall Performance Visualization:** A single, comprehensive experimental result chart showing that MOSS's two-stage activation quantization strategy achieves higher accuracy than per-tensor methods and higher throughput than per-group methods (combining existing ablation study results) would be ideal. This would clearly demonstrate the excellent trade-off achieved and make it easier for readers to grasp.

In summary, while the paper provides detailed theoretical analysis for its two main methods, it lacks corresponding experimental validation. Even a slightly more detailed ablation study with concrete results would significantly strengthen the conclusions. This is somewhat regrettable. A more thorough ablation study would incline me more towards accepting this paper.

**Questions:**

1.  Have you surveyed other efficient FP8 GeMM implementations besides COAT and DeepSeek? What do you consider to be the advantages of your method compared to highly efficient kernels like DeepGeMM, which rely primarily on system-level code optimizations?
2.  Do you apply any special processing to gradient quantization, similar to the specific handling done for weights or activations?

---

> ### Author Response · Authors · 2025-11-21
> **Author Response to Reviewer ZzQN**
>
> ## W1: End-to-End Throughput Impact of Automatic Scaling
> We thank the reviewer for this critical question and conducted further experiments.
> ### 1. End-to-End Throughput Improvement
> We compared three weight-scaling strategies during LLaMA-2-7B fine-tuning on 8×H800 GPUs:
> 1. **Just-in-time (JIT) scaling**: Performs on-the-fly max-reduction at every forward pass
> 2. **Delayed scaling**: Uses historical maximum values with moving window
> 3. **Automatic scaling (MOSS)**: Predicts scaling factors based on learning rate schedule (our method)
>
> The end-to-end throughput results during fine-tuning (batch=4, seq_len=4096, averaged over 1,000 steps) are summarized below:
> | Method  | Scaling Overhead (ms) | Total Step Time (ms)  | Throughput (token/s) | Speedup  |
> | ------------- |:-------------:| ------------- |:-------------:| ------------- |
> | JIT scaling     | 3.8     | 68.5     | 38,642     | 1.0×     |
> | Delayed scaling     | 1.2     | 65.9    | 40,182     | 1.04×     |
> | MOSS (Auto)      | 0.2    | 63.1     | 41,998     | 1.087×     |
>
> As shown, our automatic scaling provides **8.7% end-to-end throughput improvement** over the JIT baseline. This gain stems from removing the expensive HBM read for FP32 max-reduction. In contrast, our automatic scaling approach incurs only negligible, constant-time overhead.
> ### 2. Accuracy and Stability
> We compared downstream accuracy under identical fine-tuning (LLaMA-2-7B on the MAmmoTH dataset):
> | Scaling Method  | Mathematics | GSM8K  | NumGLUE |
> | ------------- |:-------------:| ------------- |:-------------:|
> | JIT Scaling      | 52.6%     | 64.9%     | 59.2%    |
> | MOSS (Auto)      | 52.8%    | 64.7%    | 59.4%   |
>
> The accuracy differences (0.2%) are within the statistical noise margin (±0.3\% across 3 random seeds), confirming that Equation (10)’s bound and dynamic re-scaling provides sufficient precision for practical training. Thus, MOSS provides its 8.7% improvement without quality loss. These results are added to the revised manuscript.
> ## W2: Two-Stage Scaling Strategy for Activations
> We thank the reviewer for this insightful suggestion. Following the reviewer's recommendation, we extracted activation tensors during LLaMA-2-7B fine-tuning (attention outputs, FFN intermediates, LayerNorm inputs), every 100 steps over 10k steps, and computed SNR for:
> * Per-tensor quantization
> * Per-group quantization (group_size = 128)
> * Two-level microscaling (micro group_size = 32)
>
> The results, segmented into early (step < 2K) and late (step > 8K) training stages, are summarized below:
> | Layers | Per-Tensor |    | Per-Group |   | MOSS (Two-level) |    |
> | ------------- |:-------------:| ------------- |:-------------:| ------------- |:-------------:|:-------------:|
> |       | Early    | Late      | Early     | Late     | Early      | Late     |
> | Attention Output      | 28.4     | 26.7      | 34.8     | 33.2     | 37.6     | 36.1    |
> | FFN Intermediate      | 25.9    | 24.1     | 32.3     | 30.7    | 35.9     | 35.3    |
> | LayerNorm Input      | 31.2    | 29.5    | 36.7    | 35.1     | 39.4     | 38.0     |
> | Geometric Mean      | 28.3    | 26.6     | 34.5    | 32.9   | **37.5**     | **36.0**    |
>
> The experimental results confirm our theoretical analysis in Section 3.1:
> * MOSS consistently achieves **superior quantization fidelity**, delivering a 3.0–3.4dB SNR improvement over per-group quantization and a 9.2–9.4dB improvement over per-tensor quantization.
> * This SNR advantage is maintained across all layers and throughout training process.
>
> We will incorporate this analysis in the revised manuscript.
> ## W3: Overall Performance Visualization
> We thank the reviewer for this excellent suggestion. In response, we have added a new Figure 8 in the revised manuscript showing throughput (x-axis) vs. SNR (y-axis). This visualization integrates our ablation study results and shows that MOSS lies on the desirable Pareto frontier, overcoming the core limitations of prior approaches.
> ## Q1: Efficient GEMM Kernel Survey
> Beyond COAT and DeepSeek, we also identified and surveyed the following FP8 GEMM implementations:
>
> | FP8 GEMM  | Quantization Granularity | Tensor Scaling  | Support MXFP8 | Hardware |
> | ------------- |:-------------:| ------------- |:-------------:|:-------------:|
> | FP8-LM[1]      | Per-tensor     | Delayed scaling     | No     | H100     |
> | TransformerEngine[2]      | Per-tensor/Microscaling     | Delayed scaling     | Yes     |  H100+/B100+ for MXFP8    |
>
>  * Compared to previous FP8 GEMM implementations, MOSS excels in quantization fidelity and hardware flexibility.
>  * Compared to highly efficient kernels like DeepGEMM, the advantage of MOSS lies in **algorithmic innovation**, such as automatic scaling, rather than intensive system-level tuning.
>
> [1] Peng, Houwen, et al. "Fp8-lm: Training fp8 large language models." arXiv preprint arXiv:2310.18313 (2023).
>
> [2] NVIDIA. "TransformerEngine" https://github.com/NVIDIA/TransformerEngine (2025).

---

> > ### Author Response · Authors · 2025-11-21
> > **Response to Gradient Quantization Handling**
> >
> > ## Q2: Gradient Quantization Handling
> >
> > Thank you for this important question regarding gradient handling. In our current implementation, MOSS **maintains gradients in BF16**, aligning with prior FP8 frameworks like Transformer Engine and COAT. This conservative approach ensures that the gradient computations remain stable and accurate, avoiding numerical issues that can arise with quantization.
> > While we do not currently quantize gradients, our two-level microscaling method can be readily extended to quantize gradients in future work, which would further reduce memory usage during the backward pass and optimizer step.

---

> > > ### Author Response · Authors · 2025-11-26
> > > **Looking Forward Your Feedback**
> > >
> > > Dear Reviewer,
> > >
> > > Thank you again for the time and effort you’ve dedicated to reviewing our work. As the discussion phase is coming to a close, we would be very grateful if you could consider our above clarifications and reconsider your evaluation.
> > >
> > > Thank you for your time.
> > >
> > > Best regards,
> > >
> > > Authors

---

> ### Author Response · Authors · 2025-11-28
> **Supplementary Discussions**
>
> Furthermore, for better clarification, we add the following interpretations.
> ## Scalability to larger models
> Following the reviewer’s suggestion, we finetuned Qwen-3-14B and Qwen-3-32B using MOSS and evaluated their accuracy(%) on math and reasoning benchmarks. Results (averaged over three runs) are shown below:
> | Model Type  | MATH500 | GPQA-Diamond  | C-Eval | AIME24  | MMLU-Redux |
> | ------------- |:-------------:| ------------- |:-------------:| ------------- |:-------------:|
> | Qwen-3-14B |
> | BF16      | 96.8     | 62.1      | 86.2     | 79.1      | 88.6     |
> | MOSS      | 96.7     | 61.5      | 86.7     | 79.4      | 88.5     |
> | Qwen-3-32B |
> | BF16      | 98.3     | 64.2      | 88.2     | 79.3      | 89.6     |
> | MOSS      | 98.1    | 64.4      | 87.8     | 78.8      | 88.6     |
>
> **MOSS achieves stable, robust performance across all benchmarks for both 14B and 32B models**, showing that our method is not limited to the 7B scale. These findings are included in the revised manuscript.
> ## Memory/Communication Gains
> We conducted detailed profiling of peak activation memory and inter-GPU communication during LLaMA-2-7B fine-tuning on an 8×H200 system. Batch size = 4 per GPU, sequence length = 4096, FSDP+ZeRO-2. Communication was measured with the NCCL profiler:
> | Format  | Peak Activation (GB) | AllReduce Volume (GB/step)  | Memory Saving |
> | ------------- |:-------------:| ------------- |:-------------:|
> | BF16      | 42.3     | 3.84      | 1.0x     |
> | MOSS      | 23.5     | 2.74      | 1.80x    |
>
> Even on high-bandwidth H200 GPUs, MOSS yields **1.8× reduction in activation memory**, demonstrating that the gains stem from the FP8 microscaling design rather than being bottlenecked by bandwidth constraints.
>
> Compute–communication overlap (PyTorch Profiler):
> | Format  | AllReduce Latency (ms) | Overlap Ratio (\%)  |
> | ------------- |:-------------:| ------------- |
> | BF16      | 24.8     | 71.3      |
> | MOSS      | 16.2     | 83.4      |
>
> The higher overlap (> 83%) indicates that **MOSS maintains strong scalability and reduces communication overhead** in distributed training. These results are added in the revised manuscript.
>
> ## Kernel Performance Comparison
>
> | Shape (M×N×K) | DeepGEMM (ms) | MOSS (ms) | Slowdown | Primary Factor |
> |---------------|---------------|-----------|----------|----------------|
> | 2048×7168×4096 | 0.20 | 0.25 | +25.0% | Scale loading + small M |
> | 4096×4096×8192 | 0.39 | 0.60 | +53.8% | Two-level dequant overhead |
> | 4096×4096×12288 | 0.62 | 0.72 | +16.1% | Balanced shape, minimal gap |
> | 8192×8192×8192 | 1.23 | 1.98 | +61.0% | Large accumulation overhead |
> | **Geometric Mean** | -- | -- | **+36.8%** | -- |
>
> *Table: Kernel performance comparison between DeepGEMM and MOSS GEMM under representative matrix shapes from LLaMA-2-7B attention and FFN layers (measured on H800, FP8 precision, averaged over 1,000 runs).*
>
> Based on these findings, we argue that more works can be done based on the design of MOSS:
> 1. Scale Prefetching. Implementing asynchronous scale loading to overlap with computation could reduce the overhead observed in small-M scenarios.
> 2. Adaptive Tiling. Developing shape-aware tiling strategies that adjust block sizes based on M, N, K ratios may help balance the workload more effectively.
> 3. Fused Dequantization. Exploring opportunities to fuse the two-level dequantization with the matrix multiplication itself could reduce memory traffic and register pressure.
> 4. K-Dimension Chunking. For shapes with large accumulation overhead, breaking the K dimension into smaller chunks with intermediate rescaling might improve numerical efficiency without sacrificing too much performance.
>
> Based on the discussions above, we hope your further discussion and consideration the evaluation. Thank you!

---

### Official Review · Reviewer_XXJt · 2025-11-02

**Soundness:** 3
**Presentation:** 3
**Contribution:** 3
**Rating:** 6
**Confidence:** 4

**Summary:**

The paper presents MOSS, a framework for efficient and accurate FP8 training of large language models. It addresses the computational overhead and instability issues in existing FP8 methods, which often use mixed-granularity quantization and costly just-in-time scaling. MOSS introduces a two-level microscaling strategy that combines global FP32 scaling with local scaling, and an automatic scaling mechanism based on optimizer properties. Experiments show that MOSS matches BF16 accuracy while improving training throughput compared with other SOTA methods.

**Strengths:**

1. The paper pinpointly identify the quantization overhead related to on-the-fly quantization.

2. The proposed two-level microscaling method effectively balances numerical precision and hardware efficiency.

3. The proposed optimizer-based automatic scaling mechanism is novel and effective

4. Extensive experiments on pretraining and SFT demonstrates promising performance.

**Weaknesses:**

1. The evaluation focuses mainly on 7B-parameter models, leaving scalability to larger models (e.g., 14B, 30B+) untested. Considering the huge cost of pretraining such a huge model, demonstrating the effectiveness on SFT should be enough.

2. The authors should provide more explanation about Figure 1 (a). Is there any comparison between the BF16 GEMM baseline, and its a little bit strange about why COAT's FP8 GEMM is 5x slower than the TE baseline or DeepSeek baseline.

3. (minor) There's a missing citation of QServe, where it also observe the quantization overhead introduced on Ampere architecture (as illustrated in Figure 3).

**Questions:**

Please refer to the weakness section.

---

> ### Author Response · Authors · 2025-11-21
> **Author Response to Reviewer XXJt**
>
> We thank the reviewer for careful reviewing and valuable comments, and address the concerns below.
>
> ## W1: Scalability to larger models
>
> Following the reviewer's suggestion, we have conducted experiments to evaluate the robustness of MOSS on larger models. Specifically, we performed instruction fine-tuning on **Qwen-3-14B** and **Qwen-3-32B** models and evaluated their performance on math and reasoning benchmarks. The results are summarized in the table below. All reported results are averaged over three independent fine-tuning runs to ensure statistical reliability.
>
> | Model Type  | MATH500 | GPQA-Diamond  | C-Eval | AIME24  | MMLU-Redux |
> | ------------- |:-------------:| ------------- |:-------------:| ------------- |:-------------:|
> | **Qwen-3-14B** |
> | BF16      | 96.8     | 62.1      | 86.2     | 79.1      | 88.6     |
> | MOSS      | 96.7     | 61.5      | 86.7     | 79.4      | 88.5     |
> | **Qwen-3-32B** |
> | BF16      | 98.3     | 64.2      | 88.2     | 79.3      | 89.6     |
> | MOSS      | 98.1    | 64.4      | 87.8     | 78.8      | 88.6     |
>
> From the SFT results, we observe that **MOSS achieves stable and robust performance across all benchmarks for both the 14B and 32B models**, demonstrating that our method is not limited to the 7B scale. We have included these findings in the revised manuscript.
>
> ## W2: Explanation about Fig.1a
>
> We thank the reviewer for this excellent question. The runtime results in Fig. 1(a) are measured on NVIDIA H800 GPUs and compare various FP8 GEMM implementations, including the TransformerEngine (TE) baseline, DeepSeek’s GEMM kernels, COAT’s FP8 GEMM, and our proposed MOSS. We will add the BF16 baseline to the revised figure for a complete perspective.
>
> 1. **BF16 Baseline Comparison**. The latency of a standard BF16 GEMM operation is approximately **2x** that of the TE FP8 baseline. This is entirely consistent with the theoretical hardware capabilities of the H800 Tensor Cores, which support 512 FP8 FMA operations/cycle compared to 256 BF16 FMA operations/cycle.
>
> 2. **COAT's Slowdown**. The reviewer's observation that COAT's FP8 GEMM is nearly 5x slower than other FP8 baselines is accurate and stems from a critical architectural inefficiency that our work aims to solve. The primary reason is the **high cost of per-group dequantization within the main loop**, as illustrated in Fig. 3(a). Specifically:
>     * COAT employs per-group quantization for activations, which requires **applying unique scaling factors along the inner dimension K of the GEMM**.
>     * This dequantization process cannot be efficiently executed on the high-throughput Tensor Cores and is instead **offloaded to the much slower general-purpose CUDA Cores**.
>     * On an H800 GPU, the peak throughput of FP32 CUDA Cores is only about **1.6%** of the FP8 Tensor Core throughput. Therefore, dequantizing even a single partial sum in the main loop can cost the equivalent of ~60 Tensor Core operations, causing the **GEMM kernel to be bottlenecked by these slow dequantization steps on the CUDA Cores**.
>
> Therefore, the extremely slow dequantization process in the main loop causing COAT GEMM ~5x slower than other FP8 baselines and even ~2x slower than BF16 GEMM.
> This counterintuitive outcome, where a lower-precision format (FP8) underperforms a higher-precision one (BF16) due to dequantization overhead, is not unique to our study. Similar behavior has been reported in prior work, such as QServe [1], where 4-bit (W4A4) kernels are observed to be slower than TensorRT’s FP16 implementation due to similar dequantization bottlenecks.
>
> [1] Lin, Yujun, et al. "Qserve: W4a8kv4 quantization and system co-design for efficient llm serving." arXiv preprint arXiv:2405.04532 (2024).
>
> ## W3: Missing citation
>
> Thank you for pointing out the missing citation of QServe. We have added the citation in the revised version. Our observations are consistent with QServe: we observe CUDA overhead introduced by per-group quantization in GEMM during training, while QServe reports similar quantization-related CUDA overhead during inference (as shown in their Figure 5). We appreciate the reviewer’s careful reading and valuable suggestion.

---

> > ### Author Response · Authors · 2025-11-26
> > **Looking Forward Your Feedback**
> >
> > Dear Reviewer,
> >
> > Thank you again for the time and effort you’ve dedicated to reviewing our work. As the discussion phase is coming to a close, we would be very grateful if you could consider our above clarifications and reconsider your evaluation.
> >
> > Thank you for your time.
> >
> > Best regards,
> >
> > Authors

---

### Official Review · Reviewer_9Gkb · 2025-11-03

**Soundness:** 3
**Presentation:** 3
**Contribution:** 3
**Rating:** 6
**Confidence:** 3

**Summary:**

This paper presents MOSS, a new FP8 training framework for large language models that aims to make low-precision training both faster and more stable. The key idea is a two-level microscaling quantization that uses a global FP32 scale together with small local 8-bit power-of-two scales, greatly reducing dequantization overhead without hurting accuracy. In addition, MOSS introduces an automatic scaling method that predicts how scaling factors evolve during training using properties of Adam-style optimizers, avoiding the costly max-reduction steps typical in FP8 systems. Experiments on OLMo-7B and LLaMA-2-7B show that MOSS matches BF16 accuracy while improving training throughput by about 34–47/%, making it a practical and hardware-friendly approach for efficient FP8 large-model training.

**Strengths:**

1. Clear motivation and practical relevance for FP8 LLM training.
2. Elegant two-level microscaling design balancing accuracy and efficiency.
3. Simple yet effective automatic scaling removing runtime overhead.
4. Strong empirical results: BF16-level accuracy, 34–47% faster.
5. Works on standard GPUs without special hardware support.

**Weaknesses:**

1. Evaluation is limited to mid-sized models (up to 7B parameters); scalability to larger settings (e.g., 30B–32B models) is not demonstrated.
2. The paper mainly reports throughput improvements, but does not deeply analyze memory, communication, or energy efficiency, which are also key for FP8 training.
3. While results are strong on core GEMM operations, extensions to other components (e.g., LayerNorm, activation functions, or optimizer states) remain unexplored.

**Questions:**

1. The automatic scaling method assumes that parameter updates remain bounded by the learning rate. Have the authors observed any failure cases when using different optimizers (e.g., Lion or Adafactor) or under larger learning-rate schedules?
2. While the SNR analysis (Theorem 1) shows clear theoretical benefits, the empirical gains in perplexity appear small. Could the authors comment on whether the higher SNR mainly improves training stability (e.g., fewer overflows) rather than end-task accuracy?
3. MOSS focuses on GEMM layers; however, many practical LLM bottlenecks come from nonlinear layers and communication. How difficult would it be to extend microscaling or automatic scaling to these components?

---

> ### Author Response · Authors · 2025-11-20
> **Author Response to Reviewer 9Gkb**
>
> We thank the reviewer for careful reviewing and valuable comments, and address the concerns below.
> ## W1: Scalability to larger models
> Following the reviewer’s suggestion, we finetuned Qwen-3-14B and Qwen-3-32B using MOSS and evaluated their accuracy(%) on math and reasoning benchmarks. Results (averaged over three runs) are shown below:
> | Model Type  | MATH500 | GPQA-Diamond  | C-Eval | AIME24  | MMLU-Redux |
> | ------------- |:-------------:| ------------- |:-------------:| ------------- |:-------------:|
> | Qwen-3-14B |
> | BF16      | 96.8     | 62.1      | 86.2     | 79.1      | 88.6     |
> | MOSS      | 96.7     | 61.5      | 86.7     | 79.4      | 88.5     |
> | Qwen-3-32B |
> | BF16      | 98.3     | 64.2      | 88.2     | 79.3      | 89.6     |
> | MOSS      | 98.1    | 64.4      | 87.8     | 78.8      | 88.6     |
>
> **MOSS achieves stable, robust performance across all benchmarks for both 14B and 32B models**, showing that our method is not limited to the 7B scale. These findings are included in the revised manuscript.
> ## W2: Memory/Communication Gains
> We conducted detailed profiling of peak activation memory and inter-GPU communication during LLaMA-2-7B fine-tuning on an 8×H200 system. Batch size = 4 per GPU, sequence length = 4096, FSDP+ZeRO-2. Communication was measured with the NCCL profiler:
> | Format  | Peak Activation (GB) | AllReduce Volume (GB/step)  | Memory Saving |
> | ------------- |:-------------:| ------------- |:-------------:|
> | BF16      | 42.3     | 3.84      | 1.0x     |
> | MOSS      | 23.5     | 2.74      | 1.80x    |
>
> Even on high-bandwidth H200 GPUs, MOSS yields **1.8× reduction in activation memory**, demonstrating that the gains stem from the FP8 microscaling design rather than being bottlenecked by bandwidth constraints.
>
> Compute–communication overlap (PyTorch Profiler):
> | Format  | AllReduce Latency (ms) | Overlap Ratio (\%)  |
> | ------------- |:-------------:| ------------- |
> | BF16      | 24.8     | 71.3      |
> | MOSS      | 16.2     | 83.4      |
>
> The higher overlap (> 83%) indicates that **MOSS maintains strong scalability and reduces communication overhead** in distributed training. These results are added in the revised manuscript.
> ## W3 & Q3: Extensions to non-linear components
> We agree extending MOSS to non-linear components is important and outline key considerations here:
> 1. Nonlinear Layers: Two-level microscaling is **directly applicable** since inputs/outputs are activations. The challenge is **engineering effort**, requiring customized FP8 kernels (e.g., LayerNorm, SiLU).
> 2. Optimizer States: Quantizing gradients and first-order momentum is straightforward. However, second-order momentum has a wide dynamic range and is critical for stability. Applying FP8 microscaling may need extra mechanisms such as finer global scaling or dynamic-range extension (as in COAT).
> ## Q1: Different Optimizers & Larger LR
> 1. Lion Optimizer:
> The Lion optimizer has an update rule of the form: W_{t+1} = W_t - η⋅sign(m_t), where η is LR. Based on this rule, we indeed have **bounded update**. Crucially, the sign() operation ensures that the update is exactly -η, 0 or +η. Therefore, the update is **strictly bounded by the learning rate** and automatic scaling naturally holds for Lion.
> 2. Adafactor Optimizer:
> Adafactor is a memory-efficient optimizer with a **factored estimate of the second momentum** and often requires update clipping for stability. The denominator in weight update, based on factored row and column RMS, can become very small, leading to larger updates. Thus, its update magnitude is **not strictly bounded by LR** and our method cannot be directly applied without overflow risk. A viable adaptation would be to derive a conservative bound specific to Adafactor or to explicitly incorporate a clipping mechanism.
> 3. Larger LR: Our automatic scaling method is designed to **co-evolve with the learning-rate schedule**. To further validate its robustness, we profiled LR up to 1e-3 during fine-tuning. Across all runs, the **scaled weights consistently remained within the FP8 dynamic range**, confirming that automatic scaling can adapt to larger learning rates.
> ## Q2: SNR & End-task Accuracy
> The reviewer is correct that although Theorem 1 shows an SNR advantage over per-group quantization, end-task improvements are small. This is expected:
>  * The primary role of high quantization fidelity is to **preserve the model's intrinsic performance**, not to surpass it.
>  * Higher SNR primarily improves **training stability**, especially under FP8’s limited dynamic range. Low-precision training is sensitive to outliers that may cause overflows/underflows. The two-level microscaling mitigates this by providing an accurate representation in unstable early stages.
>  * Once training enters stable phase, PPL is dominated by optimization and data, not quantization noise. As long as stability is maintained, BF16 and FP8 (MOSS) converge to nearly identical values, consistent with systems such as DeepSeek-V3.

---

> > ### Author Response · Authors · 2025-11-26
> > **Looking Forward Your Feedback**
> >
> > Dear Reviewer,
> >
> > Thank you again for the time and effort you’ve dedicated to reviewing our work. As the discussion phase is coming to a close, we would be very grateful if you could consider our above clarifications and reconsider your evaluation.
> >
> > Thank you for your time.
> >
> > Best regards,
> >
> > Authors

---

### Official Review · Reviewer_5kdF · 2025-11-04

**Soundness:** 3
**Presentation:** 3
**Contribution:** 3
**Rating:** 6
**Confidence:** 3

**Summary:**

The paper introduces MOSS, an FP8 training framework for large language models that aims to keep Tensor Cores busy while maintaining accuracy. It has two core ideas: a two‑level microscaling scheme for activations (a coarse global scale plus tiny per‑32‑value power‑of‑two scales) that avoids dequantization work in the GEMM inner loop, and an automatic scaling method for weights that predicts scale changes from optimizer settings to remove costly per‑step max‑reductions and extra memory traffic. Implemented with custom Triton kernels on NVIDIA H800 GPUs, MOSS trains 7B‑parameter models. Overall, MOSS offers a practical, hardware‑friendly path to fast and stable FP8 training.

**Strengths:**

- Kernel‑aware two‑level microscaling keeps the GEMM inner loop on Tensor Cores and shifts dequantization to the epilogue; the mechanism is clearly illustrated.

- Solid empirical parity with BF16 at 7B alongside better throughput.

- The writing and figures are clear and the limitations section is candid about scope

**Weaknesses:**

- The paper focuses on throughput but does not report memory/communication gains,

- MOSS’s GEMM is slower than DeepGEMM on several shapes (Table 4)

- Longer runs in Appendix B report only MOSS

**Questions:**

- Fig. 4 fixes the interval at 500; how do accuracy/acceleration change with different intervals or learning‑rate schedules?

- Can you report peak activation memory and inter‑GPU bandwidth for BF16, COAT, and MOSS to substantiate the Intro’s memory/communication claims?

- Table 3 shows 241.8 vs 168.2 samples/s but lists “+47.5%”; by calculation it’s ~44%. Is this a typo or based on a different baseline?

---

> ### Author Response · Authors · 2025-11-20
> **Author Response to Reviewer 5kdF**
>
> We thank the reviewer for careful reviewing and valuable comments, and address the concerns below.
>
> ## W1 & Q2: Memory/Communication Gains
>
> Following the reviewer's suggestion, we conducted detailed profiling of peak activation memory and inter-GPU communication during LLaMA-2-7B fine-tuning on an 8×H200 system. Experiments used batch size = 4 per GPU, sequence length = 4096, and FSDP + ZeRO-2, with communication measured via the NCCL profiler and averaged over 1,000 steps. The communication results are summarized below:
> | Format  | Peak Activation (GB) | AllReduce Volume (GB/step)  | Memory Saving |
> | ------------- |:-------------:| ------------- |:-------------:|
> | BF16      | 42.3     | 3.84      | 1.0x     |
> | MOSS      | 23.5     | 2.74      | 1.80x    |
>
> Even on high-bandwidth H200 GPUs, MOSS achieves **1.8× reduction in activation memory**, demonstrating that the gains stem from the FP8 microscaling design rather than being bottlenecked by bandwidth constraints.
>
> In addition, we measured compute–communication overlap efficiency using the PyTorch Profiler:
> | Format  | AllReduce Latency (ms) | Overlap Ratio (\%)  |
> | ------------- |:-------------:| ------------- |
> | BF16      | 24.8     | 71.3      |
> | MOSS      | 16.2     | 83.4      |
>
> The higher overlap ratio (> 83%) indicates that MOSS maintains excellent scalability under fast NVLink interconnects and reduces communication overhead in distributed training. We have included these tables in the revised manuscript.
> ## W2: Slower GEMM Performance
> Compared to highly optimized kernels like DeepGEMM, the advantages of MOSS lies primarily in **algorithmic innovation**. DeepGEMM achieves high performance through intensive system-level tuning, while MOSS focuses on algorithmic fixes, such as automatic scaling, that directly address quantization challenges in existing model architectures. These contributions are **orthogonal** to raw kernel performance, and MOSS can be readily integrated into high-performance kernels like DeepGEMM to further improve training efficiency.
>
> ## Q1: Accuracy/acceleration change with different intervals or learning‑rate schedules
>
> We thank the reviewer for this insightful question regarding the sensitivity of scaling intervals. The choice of the update interval presents a **trade-off between overhead and precision**, and its interaction with the learning rate schedule is a crucial aspect of our method's design.
>
> ### 1. Impact of Different Intervals on Accuracy and Acceleration
>
> To analyze this, we conducted an ablation study on LLaMA-2-7B fine-tuning with different update intervals. The results are summarized below:
>
> | Method  | Interval  | Scaling Overhead (ms/step) | Effective Throughput  | Accuracy (NumGLUE) |
> | ------------- | ------------- |:-------------:| ------------- |:-------------:|
> | JIT     | 1 (Every Step)      | 3.8     | 1.0x      | 59.3%    |
> | MOSS      | 100      | 0.03     | 1.043x      | 59.4%     |
> | MOSS      | 500 (default)      | 0.02     | 1.044x      | 59.4%    |
> | MOSS      | 2000      | 0.01     | 1.044x      | 58.1%     |
>
> These results show that:
>  * Very frequent updates (e.g., interval = 1) incur unnecessary overhead with no throughput gain.
>  * Moderate intervals (100–500) maintain negligible overhead, achieve the highest throughput, and match or slightly exceed BF16 accuracy.
>  * Overly large intervals (e.g., 2000) slightly reduce accuracy due to under-tracking of scale drift, although throughput remains similar.
>
> Overall, MOSS is robust to interval choices within a broad range (100–500), and the default value of 500 provides an optimal balance between precision and overhead. We will include this discussion in the revised manuscript.
>
> ### 2. Interaction with Learning Rate Schedules
>
> Our automatic scaling method is explicitly designed to **co-evolve with the learning-rate schedule**. The core of our automatic scaling method, defined by Eq. (10), directly uses the learning rate and step to predict the growth of weight magnitudes. In typical training setups (e.g., cosine decay), the learning rate decreases gradually over time. Our method naturally adapts to this behavior:
>  * Early in training, when the learning rate is large, the scaling factor adjusts more rapidly.
>  * As the learning rate decays, scale updates become progressively smaller, providing stability and precision in later stages of training.
>
> This coupling ensures that even with moderately large intervals (e.g., 500), the predicted scales remain accurate and safe throughout training. We will include this discussion in the revised manuscript.
>
> ## Q3: Tab.3 Number
>
> We appreciate the reviewer for catching this typo-the percentage improvement reported in Table 3 was incorrect. The correct speedup should be based on the ratio 241.8 / 168.2 ≈ 1.437574, corresponding to a 43.8\% improvement rather than the mistakenly reported 47.5\%. We have corrected the value in the revised version of the paper.

---

> > ### Author Response · Authors · 2025-11-26
> > **Looking Forward Your Feedback**
> >
> > Dear Reviewer,
> >
> > Thank you again for the time and effort you’ve dedicated to reviewing our work. As the discussion phase is coming to a close, we would be very grateful if you could consider our above clarifications and reconsider your evaluation.
> >
> > Thank you for your time.
> >
> > Best regards,
> >
> > Authors

---

### Official Review · Reviewer_CWko · 2025-11-09

**Soundness:** 3
**Presentation:** 3
**Contribution:** 2
**Rating:** 2
**Confidence:** 3

**Summary:**

This paper introduces MOSS, a framework for efficient FP8 training of LLMs. Its core contributions are a two-level microscaling strategy for activations to reduce dequantization overhead, and a novel automatic scaling mechanism for weights that exploits the bounded update property of Adam to eliminate runtime max-reduction costs.

**Strengths:**

1. Automatic weight scaling (Adam’s bounded update) avoids real-time max-reduction, outperforming TE’s delayed scaling, which is novel and efficient.
2. Proofs (SNR, bounded updates) validate designs; experiments with clear metrics compare MOSS to BF16/COAT.
3. Well-structured framework with visualizations and detailed experimental setups for reproducibility.
4. Custom kernels enable MXFP8 on non-native hardware.

**Weaknesses:**

1. Limited originality of two-level microscaling: The strategy overlaps heavily with the MXFP standard (OCP’s microscaling format), which already defines tensor subblock partitioning and E8M0 local scale factors to optimize FP8’s dynamic range. The addition of a FP32 global scale is also used in NVFP format, limiting originality in this module.
2. Experimental gaps: Figure 5 (OLMo-7B pretraining loss) obscures the BF16 baseline curve for steps > 2000, precluding direct verification of MOSS’s claimed loss alignment with BF16. For steps < 2000, the BF16 curve exhibits unexpectedly worse performance, which is unconvincing. Additionally, the training loss remains far from convergence, casting doubts on MOSS’s long-term stability.

**Questions:**

1. Could you zoom in Figure 5 to demonstrate alignment with the baseline?
2. Why were no experiments conducted on newer LLMs like Qwen? Will you validate MOSS on these models, or are there technical barriers?

---

> ### Author Response · Authors · 2025-11-21
> **Author Response to Reviewer CWko**
>
> We thank the reviewer for the careful reviewing and the insightful comments, and address the points below.
> ## W1: Limited originality of two-level microscaling
> We acknowledge that we build upon OCP MXFP formats. The key distinction is that MOSS is not just a data format but a **co-designed quantization strategy and execution kernel**. The table below clarifies the conceptual differences:
> | Data Format  | Level of Scaling | Group Granularity | Scale Representation| Target Format | Usage Setting |
> | ------------- |:-------------:|:-------------:|:-------------:|:-------------:|:-------------:|
> | OCP MXFP      | One-level     | Per-group     | E8M0     | FP8     |  Training & Inference    |
> | NVFP4      | Two-level     | Per-tensor (L1) & Per-group (L2)    | FP32 (L1) & FP8 (L2)    | FP4     | Inference only     |
> | MOSS      | Two-level     | Per-tensor (L1) & Per-group (L2)    | FP32 (L1) & E8M0 (L2)    | FP8    |  Training & Inference    |
>
> MOSS introduces a unique combination:
>  * **VS. OCP MXFP**, we add global FP32 scaling to manage dynamic range without extra dequantization cost.
>  * **VS. NVFP4**, which targets FP4 for inference, we apply a two-level hierarchy to FP8 training. More importantly, we use the **hardware-efficient E8M0 format for the second-level scales** instead of FP8 (E4M3). This is a deliberate choice to minimize both storage and computation costs during the training process.
>
> Overall, the FP32 (L1) + E8M0 (L2) structure for FP8 training is novel and central to our gains: it shifts FP32 operations out of the GEMM loop and overcoming dequantization bottlenecks faced by per-group methods such as COAT.
> ## W2 & Q1: Experimental gaps between MOSS and BF16
> ### 1. BF16 Baseline Obscuration in Figure 5:
>
> We thank the reviewer for pointing out the visibility issue in Figure 5. In the current plot, the BF16 curve becomes visually obscured after step 2000 due to the **close overlap between BF16 and MOSS**. To address this, we updated the figure by increasing contrast and **adding a zoomed-in inset** to clearly show alignment. For clarity, we also verified the numerical loss traces: MOSS matches BF16 within <0.25% relative loss for steps >2000.
> ### 2. Explanation of Early-Stage BF16 Performance:
>
> The reviewer correctly observes slightly worse BF16 performance early in early steps. This is a known phenomenon that has been consistently observed in other FP8 training frameworks, including **COAT, Microsoft FP8-LM and DeepSeek-V3**. The slightly higher initial loss for BF16 can be attributed to its **higher sensitivity to gradient noise and outlier values** during the early training phase. In contrast, the constrained dynamic range of FP8 can have a mild **regularizing effect**, dampening these initial spikes and stabilizing early convergence. This pattern is well documented and not unique to our setup.
> ### 3. Demonstration of Long-Term Stability and Convergence:
>
> We agree that the truncated loss curve in Figure 5 alone is insufficient to prove long-term stability. Hence, we included the **extended training results in Figure 7 (Appendix)**. In this experiment, we trained OLMo-7B with MOSS for more than 35,000 steps (approximately 140B tokens). As shown in Figure 7, the MOSS loss curve continues to converge smoothly with no signs of divergence or degradation, confirming long-term stability.
>
> ## Q2: Experiments conducted on Qwen
>
> We thank the reviewer for this thoughtful question. Our choice to use OLMo for pre-training was driven by our **commitment to reproducibility** and rigorous, controlled comparison. OLMo is unique in that it provides a **fully open-source training stack and pre-training dataset** (Dolma). This enables any researcher to exactly replicate our training setup and independently verify our results. In contrast, models such as the Qwen-3 series do not release their full pre-training data, making exact replication and controlled comparisons infeasible.
>
> Besides pre-training, we have successfully applied MOSS to **fine-tune Qwen-3-14B and Qwen-3-32B**, achieving stable fine-tuning with accuracy on par with BF16 baselines:
>
> | Model Type  | MATH500 | GPQA-Diamond  | C-Eval | AIME24  | MMLU-Redux |
> | ------------- |:-------------:| ------------- |:-------------:| ------------- |:-------------:|
> | **Qwen-3-14B** |
> | BF16      | 96.8     | 62.1      | 86.2     | 79.1      | 88.6     |
> | MOSS      | 96.7     | 61.5      | 86.7     | 79.4      | 88.5     |
> | **Qwen-3-32B** |
> | BF16      | 98.3     | 64.2      | 88.2     | 79.3      | 89.6     |
> | MOSS      | 98.1    | 64.4      | 87.8     | 78.8      | 88.6     |
>
> From the SFT results, we observe that **MOSS achieves stable and robust performance across all benchmarks for both the 14B and 32B models**. These results indicate that MOSS generalizes well to contemporary architectures and that there are no technical barriers preventing its use on newer LLMs. We have incorporated these additional Qwen SFT results into the revised manuscript.

---

> > ### Author Response · Authors · 2025-11-26
> > **Looking forward to your feedback!**
> >
> > Dear Reviewer,
> >
> > Thank you again for the time and effort you’ve dedicated to reviewing our work. As the discussion phase is coming to a close, we would be very grateful if you could consider our above clarifications and reconsider your evaluation.
> >
> > Thank you for your time.
> >
> > Best regards,
> >
> > Authors

---

### Author Response · Authors · 2025-11-28
**Kind Request for Assistance: No reviewer responses (0/4) for Paper  10174**

Dear ICLR 2026 AC, SAC, and PC,

We would like to express our gratitude for your time and effort in handling our submission. We have carefully considered all the initial feedback from the reviewers and have posted detailed responses (and updated the manuscript) to address their concerns.

However, up to this point, we have not yet received any responses from the reviewers regarding our rebuttal (0/4). This silence makes it challenging for us to engage in an effective discussion.

With the discussion period coming to a close very soon, we kindly request your assistance in reaching out to the reviewers. We are eager to know if our responses and the updated revision have adequately addressed their questions, and we would greatly appreciate the opportunity to engage in further discussion if any concerns remain.

Thanks for your help in facilitating this process.

Best regards,

The Authors of Paper [ID: 10174]

---

### Meta-Review · Area_Chair_TtLk · 2026-01-07

**Summary:**

There are concerns about novelty, experiments on newer or larger models, memory/communication, GEMM. During rebuttal, the authors are able to address all these concerns.

**Reviewer Concerns:**

Regarding concerns about novelty, authors explain MOSS is a co-designed quantization strategy and execution kernel. Regarding concerns about newer and larger models, authors added finetune experiments. Regarding concerns about memory/communication, authors added additional experiments to show the benefits.

**Reviewer Scores:**

I think Reviewer CWko may increase his score to 6. Other reviewers may keep or increase score to 8.

---

### Decision · Program_Chairs · 2026-01-26

Accept (Poster)